# Development of a Novel Enzyme-Linked Immunosorbent Assay and Lateral Flow Test System for Improved Serodiagnosis of Visceral Leishmaniasis in Different Areas of Endemicity

Rouzbeh Mahdavi,[a] Hosam Shams-Eldin,[a] Sandra Witt,[a] Andreas Latz,[b] Daniela Heinz,[b] Alba Fresco-Taboada,[c] Cristina Aira,[c] Marc P. Hübner,[d,e] Dalia Sukyte,[a] Alexander Visekruna,[a] Henrique C. Teixeira,[f] Elfadil Abass,[g] Ulrich Steinhoff[a]

[a]Institute of Medical Microbiology and Hospital Hygiene, Philipps University Marburg, Marburg, Germany
[b]Novatec Immundiagnostica GmbH, part of Gold Standard Diagnostics Europe, Dietzenbach, Germany
[c]Eurofins-Immunologia y Genética Aplicada S.A. (Eurofins Ingenasa S.A.), Madrid, Spain
[d]Institute for Medical Microbiology, Immunology and Parasitology, University Hospital Bonn, Bonn, Germany
[e]German Center for Infection Research (DZIF), Partner Site Bonn-Cologne, Bonn, Germany
[f]Department of Parasitology, Microbiology and Immunology, Universidade Federal de Juiz de Fora, Juiz de Fora, Brazil
[g]Department of Clinical Laboratory Science, College of Applied Medical Sciences, Imam Abdulrahman Bin Faisal University, Dammam, Saudi Arabia

**ABSTRACT** Visceral leishmaniasis (VL) is caused by protozoan parasites of the *Leishmania donovani* complex and is one of the most prominent vector-borne infectious diseases with epidemic and mortality potential if not correctly diagnosed and treated. East African countries suffer from a very high incidence of VL, and although several diagnostic tests are available for VL, diagnosis continues to represent a big challenge in these countries due to the lack of sensitivity and specificity of current serological tools. Based on bioinformatic analysis, a new recombinant kinesin antigen from *Leishmania infantum* (rKLi8.3) was developed. The diagnostic performance of rKLi8.3 was evaluated by enzyme-linked immunosorbent assay (ELISA) and lateral flow test (LFT) on a panel of sera from Sudanese, Indian, and South American patients diagnosed with VL or other diseases, including tuberculosis, malaria, and trypanosomiasis. The diagnostic accuracy of rKLi8.3 was compared with rK39 and rKLO8 antigens. The VL-specific sensitivity of rK39, rKLO8, and rKLi8.3 ranged from 91.2% over 92.4% to 97.1% and specificity ranged from 93.6% over 97.6% to 99.2%, respectively. In India, all tests showed a comparable specificity of 90.9%, while the sensitivity ranged from 94.7% to 100% (rKLi8.3). In contrast to commercial serodiagnostic tests, rKLi8.3-based ELISA and LFT showed improved sensitivity and no cross-reactivity with other parasitic diseases. Thus, rKLi8.3-based ELISA and LFT offer improved VL serodiagnostic efficiency in East Africa and other areas of endemicity.

**IMPORTANCE** Reliable and field suitable serodiagnosis of visceral leishmaniasis (VL) in East Africa has until now been a big challenge due to low sensitivity and cross-reactivity with other pathogens. To improve VL serodiagnosis, a new recombinant kinesin antigen from *Leishmania infantum* (rKLi8.3) was developed and tested with a panel of sera from Sudanese, Indian, and South American patients diagnosed with VL or other infectious diseases. Both prototype rKLi8.3-based enzyme-linked immunosorbent assay (ELISA) and lateral flow test (LFT) showed improved sensitivity and no cross-reactivity with other parasitic diseases. Thus, rKLi8.3-based ELISA and LFT offer substantially increased diagnostic efficiency for VL in East Africa and other areas of endemicity, compared to currently commercially available serodiagnostic tests.

**KEYWORDS** visceral leishmaniasis, serodiagnosis, VL in East Africa, lateral flow assay, ELISA, improved kinesin antigen

Address correspondence to Ulrich Steinhoff, ulrich.steinhoff@staff.uni-marburg.de.

The authors declare a conflict of interest. U.S. and R.M. are inventors on a patent application related to the use of rKLi8.3 that has been filed by the Philipps-University Marburg (EP22152398.8). The title of patent application: Diagnostic test for high sensitive detection of antibodies from visceral Leishmaniasis patients. A.L. and D.H. are employees of NovaTec Immundiagnostica GmbH, A.F.-T. and C.A. are employees of Eurofins Ingenasa S.A. and were involved in the development of prototype ELISA and Lateral flow tests.

*[This article was published on 19 April 2023 with Henrique C. Teixeira's name misspelled as "Teixera" in the byline. The byline was corrected in the current version, posted on 24 April 2023.]*

Diseases caused by the kinetoplastid parasites, including *Trypanosoma brucei*, *Trypanosoma cruzi*, and *Leishmania* spp., affect 20 million people worldwide and result in more than 50,000 deaths annually (1). Visceral leishmaniasis (VL) is the most severe disease mainly caused by *Leishmania donovani* and *Leishmania infantum* and it is transmitted to humans by the bite of the sandfly vector. Thus, for effective prevention and disease control, early diagnosis and effective treatment of VL is necessary to reduce disabilities and death of infected patients. In addition, screening of potential animal reservoirs, i.e., dogs, that are in close contact to humans, is another effective measure to reduce the potential of VL transmission. Within the *L. donovani* complex, different *Leishmania* spp. are responsible for VL in East Africa, India, and South America (2). Limited access to medical facilities and clinical symptoms of VL that are shared with other diseases, such as malaria, typhoid fever, schistosomiasis, amebic liver abscess, trypanosomiasis, and tuberculosis, require field-suitable and accurate diagnostic tests before treating patients (3). Diagnosis often relies on parasitological and serological examinations (4). Direct parasitological diagnosis is often invasive as it relies on the visualization of amastigote forms in smears obtained from aspirates of infected tissue. In contrast, serological tests are simple to perform and allow the detection of anti-*Leishmania* antibodies in the serum, e.g., by direct agglutination test (DAT) (5), enzyme-linked immunosorbent assay (ELISA), or lateral flow test (LFT). Notwithstanding, the sensitivity of serologic assays may be compromised by the heterogeneity of *L. donovani* parasites even in the same region of endemicity.

During the course of VL infection, most patients develop a strong antibody response against epitopes of the kinesin, a motor protein of the microtubule cytoskeleton involved in the growth and differentiation of *Leishmania* and other parasites (6, 7). Kinesins are composed of tandem repeats (TRs), and proteins containing such repetitive domains are found across all kingdoms of life and are often potent T- and B-cell antigens. VL patients develop strong antibody responses against kinesin, and thus, these proteins are suitable antigens for VL diagnosis (8–10).

We have previously shown that the recombinant protein rKLO8, derived from a *L. donovani* strain from Sudan (2), provides good sensitivity for VL diagnosis in East Africa. However, the high variability of kinesins even among VL strains from the same area of endemicity (11–14) suggests that the high performance of VL serodiagnosis not only relies on a particular kinesin sequence but also on the quantity of B-cell epitopes within a diagnostic antigen. We here tested rKLi8.3, a novel multiepitope-containing antigen, in the format of ELISA and LFT for their potential to improve VL diagnosis in Sudan, India, and Brazil.

## RESULTS

**Selection of an appropriate kinesin antigen to improve VL serodiagnosis.** Kinesins of VL strains from Sudan were sequenced and compared with published sequences of KE16 (India), rK39 (Brazil), and rKLO8 (Sudan). A wide genetic diversity of kinesins between VL strains from Sudan, India, and Brazil was observed and revealed that variations were mostly associated with the origin of the parasite. Amino acid (AA) variations in the first part of TR (AAs 1 to 18) were often associated with a substitution of charged for uncharged AAs, while in the second part (AAs 18 to 39), substitutions were highly conserved between different VL isolates (Table S1).

We first assessed the VL serodiagnostic performance of kinesin antigens with different intrarepeat variations but an identical number of TRs (Table S2). Three antigens, rKLi8.3 from *L. infantum*, rKLd8.3 from *L. donovani*, and rKLa8.3 from *Leishmania archibaldi*, were tested on a panel of sera from Sudanese patients with VL ($n = 172$), malaria ($n = 5$), and tuberculosis ($n = 26$), as well as endemic ($n = 85$) and nonendemic ($n = 10$) healthy controls. All antigens showed the same specificity (99.2%), but rKLi8.3 revealed the highest sensitivity (97.1%) and diagnostic efficiency value (97.98%) (Fig. 1; Table 1).

Next, the impact of the number of TRs on VL antibody binding was analyzed. Recombinant kinesins from *L. infantum* with increasing numbers of TR (rKLi6.3, rKLi7.3, and rKLi8.3) were tested for VL antibody binding in ELISA with the same panel of sera (Fig. S2). As shown in Fig. 2 and Table 2, the sensitivity and specificity were enhanced with increasing numbers

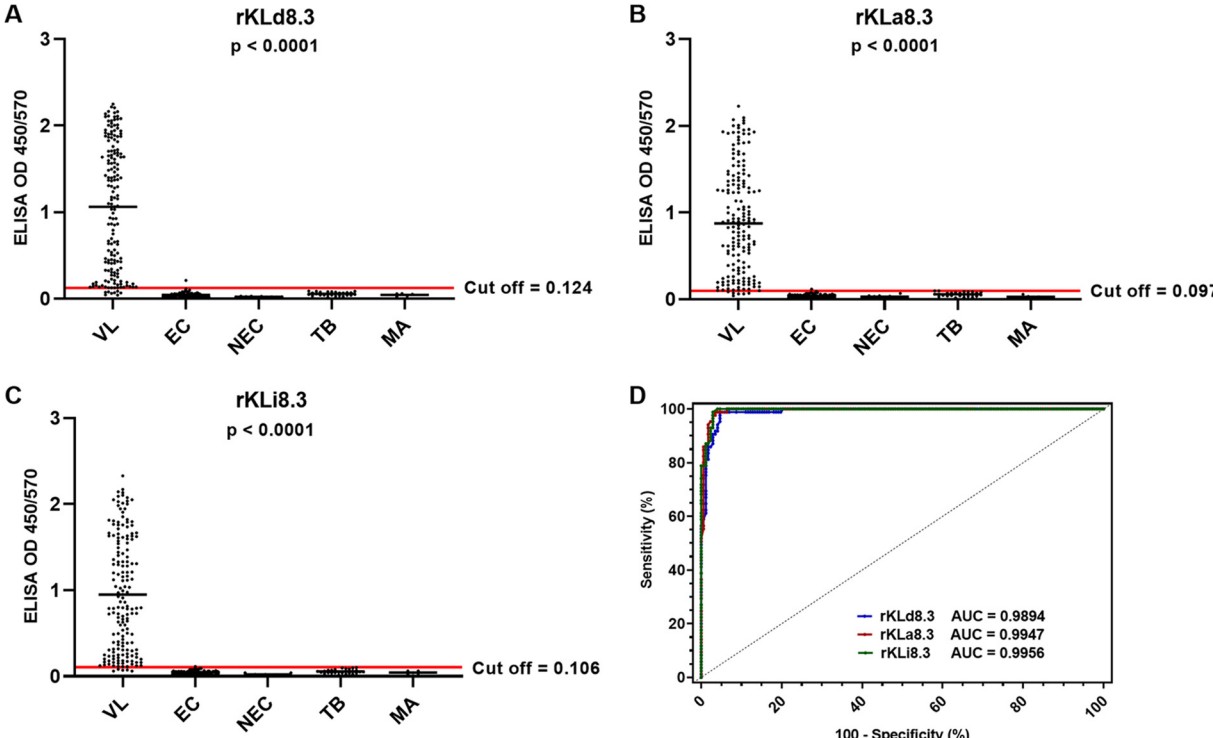

**FIG 1** Influence of intrarepeat variations on visceral leishmaniasis (VL) specific antibody responses. (A to C) rKLd8.3-based (A), rKLa8.3-based (B), and rKLi8.3-based (C) enzyme-linked immunosorbent assay (ELISA) in VL (*n* = 172), endemic controls (EC) (*n* = 85), nonendemic controls (NEC) (*n* = 10), tuberculosis (TB) (*n* = 26), and malaria (MA) (*n* = 5). (D) Receiver operating characteristic (ROC) curve of rKLd8.3-, rKLa8.3-, and rKLi8.3-based ELISA. AUC, area under the curve; OD, optical density.

of TR from 92.44% to 97.10% and from 96.82% to 99.20%, respectively. Most strikingly, false-negative samples decreased from 13 (rKLi6.3) to 5 (rKLi8.3). Receiver operating characteristic (ROC) curve analysis confirmed the best diagnostic performance with rKLi8.3, followed by rKLi7.3 and rKLi6.3. In addition, bioinformatic analyses of linear B-cell epitopes of kinesins with different number of TR confirmed that rKLi8.3 has the highest score for B-cell antigenicity (Fig. S3).

**Comparative testing of rK39, rKLO8, and rKLi8.3 antigens for VL serodiagnosis in Sudan and India.** To assess whether rKLi8.3 improves VL serodiagnosis, we compared the performance of rKLi8.3 with our previously developed rKLO8 and the commercially available rK39 in ELISA with the above-mentioned panel of sera from Sudanese patients. The rK39 ELISA showed a sensitivity of 91.28% and a specificity of 93.65%, respectively. While the rKLO8 ELISA showed similar sensitivity (92.44%) but higher specificity (97.62%), the sensitivity and specificity of rKLi8.3 were 97.1% and 99.2%, respectively (Fig. 3). The number of false negatives (FNs) declined from 15 (rK39) to 5 (rKLi8.3), and the number of false positives (FPs) declined from 8 (rK39) to 1 (rKLi8.3) (Table 3). ROC curve analysis confirmed that diagnostic efficiency increased from rK39 to rKLO8 and rKLi8.3 with 92.28%, 94.63%, and 97.98%, respectively.

In addition, sera from Indian VL patients (*n* = 19) and endemic controls (*n* = 11) were tested. A similar specificity for all three antigens but increased sensitivity from 94.73% (rK39 and rKLO8) to 100% (rKLi8.3) was observed (Fig. S4; Table S3).

**TABLE 1** VL-diagnostic performance of rKLd8.3, rKLa8.3, and rKLi8.3 ELISA in Sudan[a]

| ELISA test | Cutoff | AUC | TP | FN | TN | FP | Sensitivity (%) | Specificity (%) | PPV (%) | NPV (%) | DEV (%) |
|---|---|---|---|---|---|---|---|---|---|---|---|
| rKLd8.3 | 0.124 | 0.9894 | 164 | 8 | 125 | 1 | 95.35 | 99.20 | 99.39 | 93.98 | 96.97 |
| rKLa8.3 | 0.097 | 0.9947 | 166 | 6 | 125 | 1 | 96.51 | 99.20 | 99.40 | 95.41 | 97.65 |
| rKLi8.3 | 0.106 | 0.9956 | 167 | 5 | 125 | 1 | 97.10 | 99.20 | 99.40 | 96.15 | 97.98 |

[a]AUC, area under the curve; DEV, diagnostic efficiency value; ELISA, enzyme-linked immunosorbent assay; FN, false negative; FP, false positive; NPV, negative predictive value; PPV, positive predictive value; TN, true negative; TP, true positive; VL, visceral leishmaniasis.

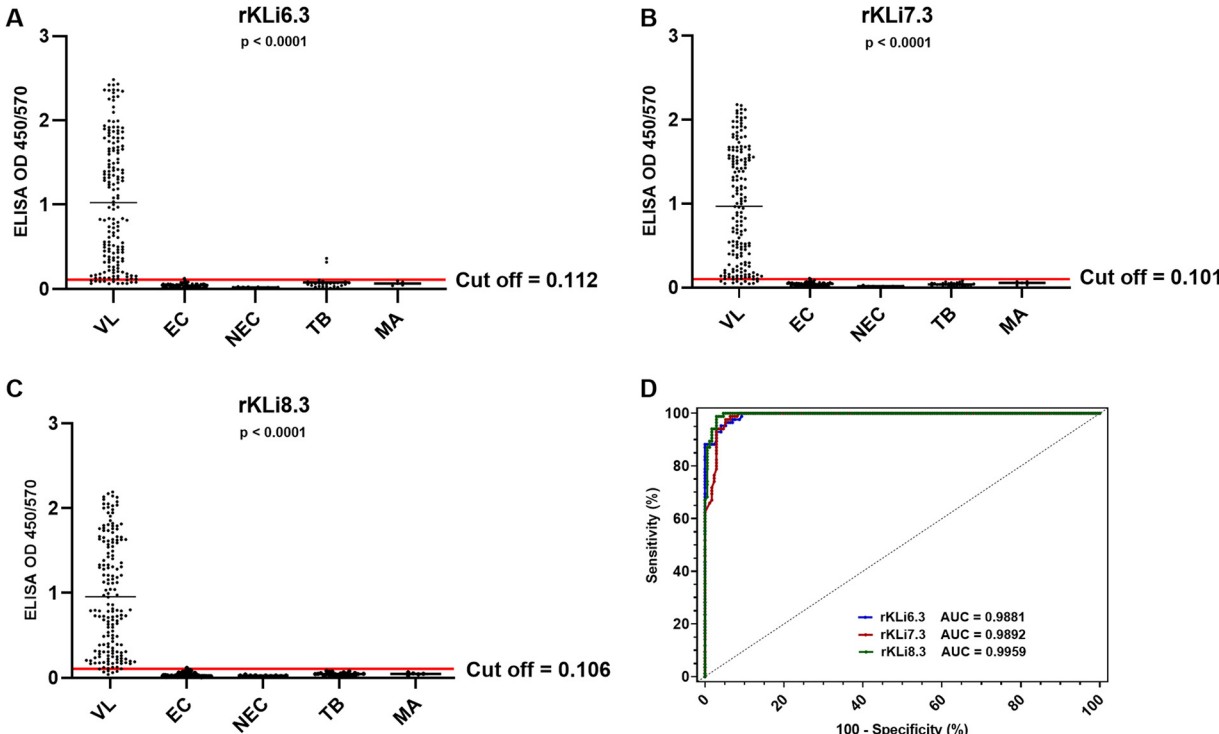

**FIG 2** Impact of the number of TRs on VL specific antibody responses. (A to C) rKLi6.3-based (A), rKLi7.3-based (B), and rKLi8.3-based (C) ELISA in VL (*n* = 172), endemic controls (EC) (*n* = 85), nonendemic controls (NEC) (*n* = 10), tuberculosis (TB) (*n* = 26), and malaria (MA) (*n* = 5). (D) ROC curve of rKLi6.3-, rKLi7.3-, and rKLi8.3-based ELISA.

**Evaluation of prototype rKLi8.3-based ELISA and lateral flow test.** The diagnostic performance of the rKLi8.3 prototype ELISA was evaluated and compared with commercially available ELISAs (TECAN/NovaLisa) on a panel of sera from patients with VL (*L. infantum*; *n* = 8), cutaneous leishmaniasis (CL) (*Leishmania major*, *Leishmania tropica*, and *Leishmania brasiliensis*; *n* = 14), and Chagas disease (*T. cruzi*; *n* = 8). All procedures were performed according to the manufacturer's instructions, and data within the borderline zone were considered negative. The sensitivity and specificity of TECAN/NovaLisa and rKLi8.3 ranged from 50% to 75% and from 72.72% to 95.45%, respectively (Table 4). ROC curve analysis confirmed that rKLi8.3 has the highest diagnostic efficiency value (DEV) with 90%, and no cross-reactivity with *T. cruzi* was observed. In contrast, three sera from *T. cruzi*-infected patients gave false-positive signals when tested with the TECAN/NovaLisa ELISA (Fig. 4). Furthermore, the rKLi8.3 prototype ELISA was tested against a panel of sera from patients infected with *Schistosoma mansoni*, *Entamoeba histolytica*, *Plasmodium falciparum*, *Toxoplasma gondii*, and *Echinoccocus multilocularis*. None of these sera showed any cross-reactivity, underlining the specificity of rKLi8.3 (data not shown).

In addition, a prototype rKLi8.3 LFT and a commercially available rK39 rapid diagnostic test (RDT) (IT Leish, Bio-Rad) were tested with the same sera used in Fig. 1. The data showed that the rKLi8.3 LFT has increased sensitivity and specificity compared to IT Leish, with 86.96% to 95.65% and 84.62% to 97.44%, respectively (Table 5). In contrast to IT Leish, none of the control sera (tuberculosis and malaria) showed cross-reaction with the rKLi8.3 LFT.

**TABLE 2** VL-diagnostic performance of rKLi6.3, rKLi7.3, and rKLi8.3 ELISA in Sudan[a]

| ELISA test | Cutoff | AUC | TP | FN | TN | FP | Sensitivity (%) | Specificity (%) | PPV (%) | NPV (%) | DEV (%) |
|---|---|---|---|---|---|---|---|---|---|---|---|
| rKLi6.3 | 0.112 | 0.9881 | 159 | 13 | 122 | 4 | 92.44 | 96.82 | 97.54 | 90.37 | 94.29 |
| rKLi7.3 | 0.101 | 0.9892 | 162 | 10 | 124 | 2 | 94.18 | 98.41 | 98.78 | 92.54 | 95.97 |
| rKLi8.3 | 0.106 | 0.9959 | 167 | 5 | 125 | 1 | 97.10 | 99.20 | 99.40 | 96.15 | 97.98 |

[a]AUC, area under the curve; DEV, diagnostic efficiency value; ELISA, enzyme-linked immunosorbent assay; FN, false negative; FP, false positive; NPV, negative predictive value; PPV, positive predictive value; TN, true negative; TP, true positive; VL, visceral leishmaniasis.

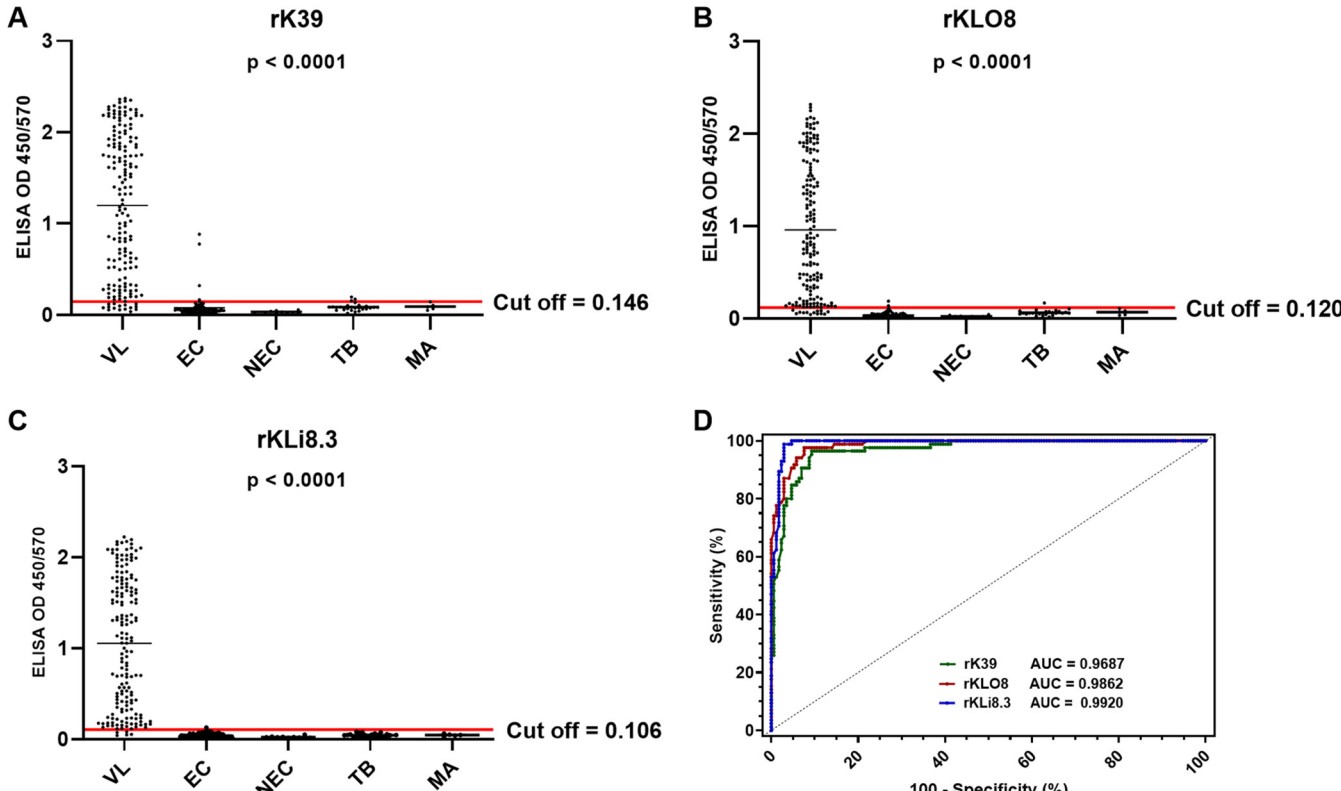

**FIG 3** (A to C) VL-specific antibody responses from patients in East Africa were measured by rK39-based (A), rKLO8-based (B), and rKLi8.3-based (C) ELISA in VL (*n* = 172), tuberculosis (TB) (*n* = 26), malaria (MA) (*n* = 5), endemic controls (EC) (*n* = 85), and nonendemic controls (NEC) (*n* = 10). (D) ROC curves for rK39-, rKLO8-, and rKLi8.3-based ELISA.

## DISCUSSION

East Africa is currently the most affected region in the world accounting for 45% of VL cases mainly due to politically and economically forced migration. The lack of a highly sensitive rapid test, as well as the complex treatment, which requires medical facilities, make VL control more challenging in East Africa than in other areas of endemicity (15). Thus, reliable diagnostic tests are urgently needed in these regions (16).

The kinesin-related antigen rK39 and KE16 (17) are currently used as rapid test format for VL serodiagnosis (18, 19). A systematic review estimated that rK39 RDT has a sensitivity of 97% in the Indian subcontinent but only 85% in East Africa (20). Intermediate values (88% to 94%) have been reported for the rK39 RDT in Brazil (21). The previously developed rK28 prototype RDT (22) showed high sensitivity but low specificity in Africa (23). The aim of this study was to develop and evaluate an improved VL-specific ELISA and RDT for East Africa and other areas of endemicity. For this purpose, we analyzed kinesin antigens of different *Leishmania* isolates with regard to the impact of sequence variations and number of TR for strong and specific VL antibody binding. Sequence analyses revealed that *Leishmania* kinesins within the same area of endemicity of East Africa varied in terms of AA composition, and many of these variations are associated with an altered charge of AA. Hydrophilic proteins that contain charged AA are more potent B-cell antigens than

**TABLE 3** VL-diagnostic performance of rK39, rKLO8, and rKLi8.3 ELISA in Sudan[a]

| ELISA test | Cutoff | AUC | TP | FN | TN | FP | Sensitivity (%) | Specificity (%) | PPV (%) | NPV (%) | DEV (%) |
|---|---|---|---|---|---|---|---|---|---|---|---|
| rK39 | 0.146 | 0.9687 | 157 | 15 | 118 | 8 | 91.28 | 93.65 | 95.15 | 88.72 | 92.28 |
| rKLO8 | 0.120 | 0.9862 | 159 | 13 | 123 | 3 | 92.44 | 97.62 | 98.15 | 90.44 | 94.63 |
| rKLi8.3 | 0.106 | 0.9920 | 167 | 5 | 125 | 1 | 97.10 | 99.20 | 99.40 | 96.15 | 97.98 |

[a]AUC, area under the curve; DEV, diagnostic efficiency value; ELISA, enzyme-linked immunosorbent assay; FN, false negative; FP, false positive; NPV, negative predictive value; PPV, positive predictive value; TN, true negative; TP, true positive; VL, visceral leishmaniasis.

**TABLE 4** Diagnostic performance of commercial ELISA (TECAN, NovaLisa) and rKLi8.3 prototype ELISA[a]

| ELISA test | Cutoff | AUC | Sensitivity (%) | Specificity (%) | PPV (%) | NPV (%) | DEV (%) |
|---|---|---|---|---|---|---|---|
| TECAN | 11 U | 0.7670 | 50.0 | 81.81 | 50.0 | 81.81 | 73.33 |
| NovaLisa | 11 U | 0.7784 | 50.0 | 72.72 | 40.0 | 80.0 | 66.66 |
| rKLi8.3 | 11 U | 0.9219 | 75.0 | 95.45 | 85.71 | 91.30 | 90.0 |

[a]AUC, area under the curve; DEV, diagnostic efficiency value; ELISA, enzyme-linked immunosorbent assay; NPV, negative predictive value; PPV, positive predictive value.

hydrophobic proteins. Thus, we assume that the antigenicity of kinesin is disturbed if charged AAs are substituted for uncharged AAs, which may result in decreased antibody binding (24). In addition, the copy number of TRs is also known to affect the antigen-antibody binding, because of thermodynamic binding kinetics (9). Bioinformatic analyses of linear B-cell epitopes together with our ELISA data confirmed that increasing the number of kinesin TR led to enhanced affinity of antibody binding to kinesin. Thus, we selected a conserved kinesin sequence of *L. infantum* from Sudan containing charged AAs and 8.3 TRs, both known to increase linear B-cell epitopes (25).

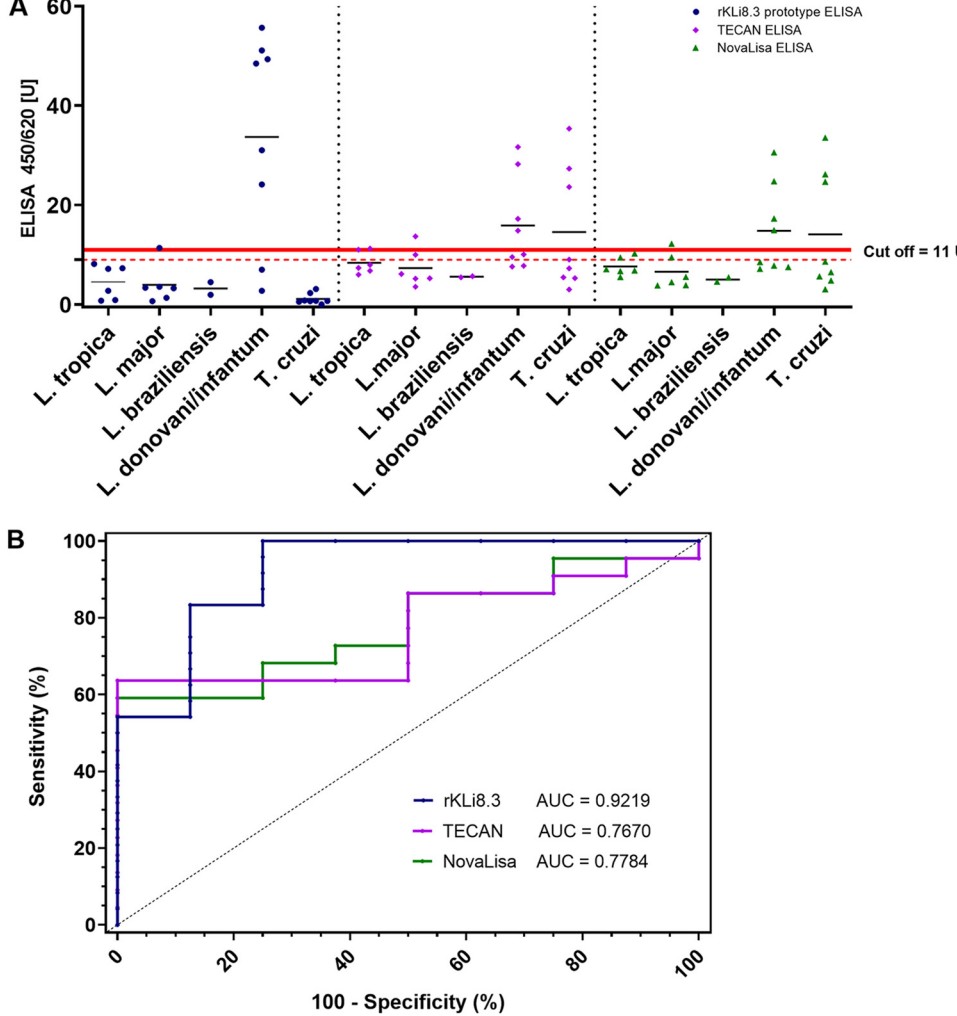

**FIG 4** Assessment of VL-specific antibodies by commercial (TECAN, NovaLisa) and rKLi8.3 prototype ELISA. (A) Sera from VL patients infected with *L. donovani/L. infantum* (*n* = 8); cutaneous leishmaniasis (CL) patients infected with *L. tropica* (*n* = 6), *L. major* (*n* = 6), *L. brasiliensis* (*n* = 2); and Chagas disease patients infected with *T. cruzi* (*n* = 8) were tested. The manufacturer-defined borderline zone is indicated by red lines; data within were considered negative. (B) ROC curve analysis of TECAN, NovaLisa, and rKLi8.3 prototype ELISA.

**TABLE 5** Performance of lateral flow tests[a]

| LFT | Manufacturer | Antigen | Sensitivity (%) | Specificity (%) | PPV (%) | NPV (%) | DEV (%) |
|---|---|---|---|---|---|---|---|
| IT Leish | Bio-Rad Laboratories, Inc. | rK39 | 86.96 | 84.62 | 76.92 | 91.67 | 85.48 |
| rKLi8.3 | Eurofins Ingenasa S.A. | rKLi8.3 | 95.65 | 97.44 | 95.65 | 97.44 | 96.77 |

[a]DEV, diagnostic efficiency value; LFT, lateral flow test; NPV, negative predictive value; PPV, positive predictive value.

The rKLi8.3 ELISA showed increased sensitivity and specificity in different VL areas of endemicity compared to the rK39 and rKLO8 ELISA and was able to reduce the number of FN and FP. Comparing the diagnostic performance of rKLi8.3 prototype with commercial ELISA kits revealed that rKLi8.3 has a better discriminatory potential between CL and VL.

Thus, the rKLi8.3 prototype ELISA exceeded TECAN/NovaLisa in terms of sensitivity and specificity. As VL patients often suffer from coinfection and comorbidities, reliable, specific, and sensitive diagnostic tools are a prerequisite for control and elimination of this disease. Sera from malaria, tuberculosis, and Chagas patients have been shown to cross-react with tests using rK39 or whole parasite lysates (22, 26–28). This may be due to shared antigenic determinants or a close phylogenetic relationship of, e.g., *Leishmania* with trypanosomes, which both belong to the family of kinetoplastida (29). In contrast, the rKLi8.3 ELISA and LFT showed no cross-reactivity with sera from tuberculosis, malaria, and Chagas patients. The high specificity of rKLi8.3 is probably due to the combination of a conserved kinesin sequence with high antigenicity (charged AAs) together with an increased number of TRs.

We are aware that the number of control sera from patients suffering from other infections was limited, as it was difficult to receive such sera from the same areas of endemicity. Despite these limitations, we were able to test sera from infections caused by 11 different pathogens, and none of these sera showed cross-reactivity with rKLi8.3. In conclusion, both rKLi8.3 ELISA and LFT provide rigorous serodiagnosis of VL patients and thus might help to improve patient care and prevent VL epidemics, especially in rural areas of East Africa.

## MATERIALS AND METHODS

**Ethics statement.** The serum samples of this study have been used in former studies that have been approved and reviewed by institutional ethics committees in the respective countries as previously stated. Serum samples have been collected with the verbal consent of the patients and have been approved by the Ethical Review Committee of the Federal Ministry of Health in Sudan (23-06-2005).

**Patients and serum sample collection.** VL diagnosis was performed by the detection of *Leishmania* parasites (amastigotes) in Giemsa-stained aspirates of lymph nodes (Doka Village, Sudan) and spleens (Bihar, India). VL suspects were based on clinical symptoms, including fever, hepatosplenomegaly, and lymphadenopathy, according to the Sudanese Ministry of Health Guidelines. All sera of confirmed VL cases were screened for HIV infection according to guidelines of Sudan National AIDS and STI control program of the Federal Ministry of Health and found to be negative. HIV testing was performed using HIV 1 and 2 simple/rapid diagnostic tests (Biorex Diagnostics, UK) and a fourth-generation HIV ELISA (1 + 2) Ag/Ab ELISA kit (Fortress Diagnostic, UK) (30). Sera from Sudan included VL patients ($n = 172$), endemic healthy controls ($n = 85$), and patients with confirmed malaria ($n = 5$) and tuberculosis ($n = 26$). Patients from India with confirmed VL ($n = 19$), malaria ($n = 1$), toxoplasmosis ($n = 1$), and healthy controls ($n = 9$) from areas of endemicity were included. Malaria was diagnosed by parasites in blood smears and tuberculosis by acid-fast bacilli in sputum smears.

Clinically and immunologically characterized sera of Brazilian patients with VL or other infections were provided by the Institute for Microbiology and Parasitology, University Hospital Bonn. Sera included patients with VL (*L. infantum*; $n = 8$), CL (*L. major, L. tropica*, and *L. brasiliensis*; $n = 14$), Chagas disease ($n = 8$), schistosomiasis ($n = 7$), amebiasis ($n = 2$), echinococcosis ($n = 3$), trichinosis ($n = 3$), malaria ($n = 5$), and toxoplasmosis ($n = 19$). In addition, sera from 10 nonendemic controls (NECs) from Germany were included in the tests. All sera were stored at $-80°C$ until use.

**Characterization and expression of recombinant kinesins.** Kinesins are composed of a varying number of tandem repeats (TRs) with 39 amino acids (AAs) each that exhibit a different degree of conservation (18). To determine the variability, kinesin genes with different numbers of TRs (6.3, 7.3, and 8.3) from East African *L. donovani* (MHOM/SD/90/D56, IORI/SD/91/D43), *L. archibaldi* (MHOM/SD/97/LEM3463, IORI/SD/91/D27, MHOM/SD/97/LEM3475), and *L. infantum* (MHOM/SD/82/GILANI) were cloned, sequenced, and analyzed for intra- and interspecific AA variations by using BLAST-NCBI and Clustal-Omega-Tool. The kinesin sequences of East African VL strains were compared with the published sequences of rK39 (GenBank accession number L07879.1), KE16 (GenBank accession number AY615886.1), and rKLO8 (GenBank accession number KC788285).

A gene fragment encoding 6.3, 7.3, and 8.3 TR (KLi6.3, KLi7.3, and KLi8.3) of a Sudanese *L. infantum* strain (MHOM/SD/82/GILANI) was amplified from promastigote genomic DNA by PCR with forward

(5′-GAGCTCGCAACCGAGTGGGAGG-3′) and reverse (5′-GCTCCGCAGCGCGCTCC-3′) primers, designed according to the published *L. infantum* JPCM5 putative kinesin K39 (GenBank accession number XM_001464261.2). Amplification products containing 6.3 to 8.3 TRs were gel purified and cloned into the plasmid vector pCR 2.1-TOPO (Invitrogen Life Technologies, USA). Competent *Escherichia coli* HB101 (Promega, Germany) were transformed with recombinant plasmids and subcloned into the pET28a(+) expression vector. The plasmids were transformed into BL21(DE3) *E. coli* cells (Sigma-Aldrich, Germany), and bacterial lysates were purified by $Ni^{2+}$ affinity chromatography and ÄKTA Prime (GE Healthcare, USA). The impact of the number of TR on B-cell antigenicity was analyzed using the prediction program of linear B-cell epitopes, BepiPred 1.0 (31).

**Recombinant antigens rK39 and rKLO8.** The rK39 antigen of *L. infantum* (*L. chagasi*) was purchased from Rekom Biotech, S.L., Granada Spain as a $6\times$ His-tagged fusion protein. The cloning, expression, and purification of the $6\times$ His-tagged fusion protein rKLO8 have recently been described (2). The concentration of proteins was verified by Bradford assay, and aliquots were kept at $-80°C$.

**Enzyme-linked immunosorbent assay.** MaxiSorp ELISA plates (Nunc Serving Life Science, Denmark) were coated with antigens (5 ng/well) in 0.1 M $NaCO_3$ buffer (pH 9.6) and incubated overnight at 4°C. The plates were washed with phosphate-buffered saline (PBS) containing 0.05% (vol/vol) Tween 20 and blocked with 3% (wt/vol) bovine serum albumin (BSA) at room temperature (RT) for 1 h. After additional washing steps, 50 $\mu$L of 1:800 diluted serum samples were added to each well and incubated for 45 min at RT. After washing, 50 $\mu$L/well of peroxidase-conjugated AffiniPure donkey anti-human IgG (H+L) (Jackson Immunoresearch Laboratories, USA) (1:10,000) was added and incubated at RT for 1 h. The reaction was activated with hydrogen peroxide, developed with tetramethylbenzidine (R&D Systems, USA), and stopped with 2 N sulfuric acid after 10 min of incubation in the dark. Optical density (OD) was measured at 450/570 nm using an ELISA microreader (FLUOstar Omega, BMG Labtech). All ELISAs have been repeated twice with serum samples in duplicate.

**Prototype lateral flow assay.** Protein A/G was used as test line reagent diluted in 20 mM Tris-HCl buffer (pH 7.5) containing sucrose and sodium azide. As control line capture reagent, a monoclonal antibody against a control protein was used. Both mixtures were dispensed in two parallel lines onto a nitrocellulose membrane using a dispense platform (Matrix 1600, Kinematic Automation). The membranes were dried at 45°C for 5 min, sealed, and stored at RT under dry conditions. The rKLi8.3 antigen and a control protein were coupled to 40-nm colloidal gold nanoparticles (AuNPs) as test and control detector reagents, respectively. AuNPs were diluted in borate buffer (pH 8.0), and target proteins were added to the mixture in the same buffer and incubated with the target/control protein for 30 min. Target and control AuNPs were mixed to a final concentration of 5 OD each, dispensed onto the conjugate pad, dried at 45°C, and stored at RT. Lateral flow strips were produced and assembled into cassettes (Fig. S1) by Eurofins Ingenasa S.A. (Madrid, Spain). A total of 20 $\mu$L of blood or 10 $\mu$L of serum was added to the application zone. After the sample was completely absorbed, 150 $\mu$L of running buffer (Tris-HCl, pH 7.5, NaCl, BSA, and $NaN_3$) was added, and the results were read after 10 min. According to the manufacturer's protocols, valid LF tests (strong internal control band) were performed once. If the control band was weak or absent, the LF tests were repeated.

**Statistical analyses.** Statistical analysis of mean optical densities (ODs) between two or multiple independent groups was assessed using Student's *t* test or one-way analysis of variance (ANOVA) test (GraphPad Software 9.0, San Diego, CA, USA). ROC areas of the tested recombinant proteins were evaluated by the nonparametric Wilcoxon tests, using MEDCALC 14.8.1 (MedCalc Software, Oostende, Belgium). Cutoff values for each recombinant protein were defined as mean OD values of sera from healthy controls plus three standard deviations (SDs). Furthermore, the cutoff value for each recombinant protein was confirmed using the receiver operating characteristic (ROC) curve. *P* values of $<0.05$ were considered statistically significant. Sensitivity, specificity, positive predictive value (PPV), and negative predictive value (NPV), and diagnostic efficiency value (DEV) were calculated to assess the usefulness of the assays at the 95% confidence intervals (32).

## SUPPLEMENTAL MATERIAL

Supplemental material is available online only.
**SUPPLEMENTAL FILE 1**, PDF file, 1 MB.

## ACKNOWLEDGMENTS

We thank Peter Walden for sharing his serum collection from Indian VL patients, Patrick Bastien for providing us *Leishmania* strains, and Anne Hellhund for excellent technical support.

All authors had access to the data in this study and take responsibility for integrity and accuracy of data analysis. Study concept and design: U.S., R.M., E.A., and H.T. Acquisition, analysis, or interpretation of data: All authors. Writing of the manuscript: U.S. and R.M. Critical review of manuscript and data: All authors. Performance of experiments: R.M., S.W., M.P.H., and D.S. Provision of human sera and diagnostic test systems: E.A., H.T., M.P.H., A.L., C.A., and A.F.-T.

This work was supported by the Loewe Center Druid through project C4 (U.S.) within the Hessian Excellence Program. Open Access funding provided by the Open Access Publishing Fund of Philipps-Universität Marburg with support of the Deutsche Forschungsgemeinschaft (DFG, German Research Foundation).

U.S. and R.M. are inventors on a patent application related to the use of rKLi8.3 that has been filed by the Philipps-University Marburg (EP22152398.8, Diagnostic test for high sensitive detection of antibodies from visceral leishmaniasis patients). A.L. and D.H. are employees of NovaTec Immundiagnostica GmbH. A.F.-T. and C.A. are employees of Eurofins Ingenasa S.A. and were involved in the development of prototype ELISA and lateral flow tests.

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
