## [Reviewer comments · Microbiology Spectrum]

Microbiology Spectrum

Development of a novel ELISA and lateral flow test system for improved serodiagnosis of visceral Leishmaniasis in different endemic areas

Rouzbeh Mahdavi, Hosam Shams-Eldin, Sandra Witt, Andreas Latz, Daniela Heinz, Alba Fresco-Taboada, Cristina Aira, Marc Hübner, Dalia Sukyte, Alexander Visekruna, Henrique Teixeira, Elfadil Abass, and Ulrich Steinhoff

Corresponding Author(s): Ulrich Steinhoff, Philipps Universität Marburg

Review Timeline:

Submission Date:	October 24, 2022
Editorial Decision:	February 3, 2023
Revision Received:	March 1, 2023
Editorial Decision:	March 16, 2023
Revision Received:	March 22, 2023
Accepted:	March 29, 2023

Editor: Kileen Shier

Reviewer(s): Disclosure of reviewer identity is with reference to reviewer comments included in decision letter(s). The following individuals involved in review of your submission have agreed to reveal their identity: Christopher Aaron Rice (Reviewer #2); Amer Al-Jawabbreh (Reviewer #4)

Transaction Report:

DOI: <https://doi.org/10.1128/spectrum.04338-22>

February 3, 2023

Prof. Ulrich Steinhoff
Philipps University of Marburg
Medical Microbiology and Hospital Hygiene
Marburg
Germany

Re: Spectrum04338-22 (Development of a novel ELISA and lateral flow test system for improved serodiagnosis of visceral Leishmaniasis in different endemic areas)

Dear Prof. Ulrich Steinhoff:

Link Not Available

Sincerely,

Kileen Shier

Journals Department
Reviewer comments:

Reviewer #2 (Comments for the Author):

Please find attached the major and minor comments.

Reviewer #3 (Comments for the Author):

The authors present a considerable amount of work that demonstrates the utility of a genetically engineered fragment of kinesin of *Leishmania infatum*, rKLi8.3, as a promising candidate to detect antibodies produced in human patients with visceral

leishmaniasis. The aim of the study presented herein was to develop two serological tests, ELISA and a lateral flow test, both of which show a higher sensitivity and specificity compared to current available in vitro diagnostic tests. The work and presentation are overall well written and should be of importance to researchers and medical technicians interested in the diagnostic of visceral leishmaniasis.

Major points

While this reviewer agrees with the authors conclusions that the sensitivity of the ELISA and the lateral flow test was increased compared to antigens previously used for the diagnosis of visceral leishmaniasis, there are some areas of improvement, in particular in the discussion of the data. But what this reviewer consider the weakness - not way a fatal flaw - is the small sample size of serum samples examined from patients with other infectious diseases used as controls. For almost all tests conducted, only a few serum samples from patients with malaria, schistosomiasis, Chagas disease, amoebiasis, and echinococcosis were employed. Also, the sample size of sera collected from India was very small (n= 21) as well. Below are comments for the authors to consider.

1. As highlighted in the general comments above, the number of serum samples from patients with other bacterial or parasitic infections used as further controls is very small while the number of control serum samples from healthy donors is quite large (n= 85). Despite the acquisition of serum samples is very difficult, it would be of importance to add more serum samples to evaluate the data collected. If collection of more serum samples is for any reason impossible, the authors have to point out this obvious limitation of the study, maybe in the discussion.
2. On page 6, the authors mentioned that kinesins have been used as antigens for the sero-diagnosis of visceral leishmaniasis caused by species of the *L. donovani* complex. In their previous work, they used a recombinant antigen, rKLO3 from an *L. donovani* isolate. Here, a novel antigen from *L. infantum* has been used for evaluation and optimization of the test. It is not entirely clear, why the authors decided to use a kinesin from *L. infantum*. Also, the rationale why a comparative sequence analyses of kinesins of *L. infantum* strains were conducted is not clear as well (see page 6, lines 191-196). Why they end up with the three kinesins described in the following paragraph? This should be clarified in the result.
3. To show the differences between the three selected antigens (rKLi6.3, rKLi7.3, and rKLi8.3) and to make the manuscript more reader-friendly, a separate figure showing the structural differences of these three antigens would be strongly recommend.
4. In figure 1, the authors presented data from the ELISA performed with three different antigens. It seems that the identical data collected with the rKLi8.3 antigen were also used for figure 2C and 3C. The authors should clearly indicate whether these data were used for the calculations of the data presented in figure 2D and 3D. To avoid any misunderstandings, the data obtained with rKLi8.3 should be only presented in figure 1.
5. On page 7, second paragraph, To evaluate the performance of the ELISA, two additional antigens, rK39 and rKLO8, were investigated. The origin or source of these antigens are unclear. Should the antigens genetically engineered by the authors, a full description including construction (amplification, cloning, etc.) and purification have to be included in the Materials and Methods. Should these antigens are commercially available, the source of these proteins have to be included.
6. The discussion is very short and needs improvement. A comprehensive discussion of the strengths and limitation (e.g. sample size of the controls are very small) of the study are strongly recommended as well as the future perspectives how to increase the sensitivity of the novel test systems. Also, the benefit of the developed tests described herein compared to in vitro diagnostics currently used have to be explored in a better way.
7. It is not clear to what the number of replicates of each serum sample was used in the ELISAs and LFT. This needs clarification.
8. The definition of the cut off values should be explained more clearly. It is not entirely clear why the authors define the mean absorbance values 'plus three standard deviations'. Please clarify.

Minor points

P2, line 44, wording of the sentence ... led to the development of new recombinant antigen... is scientifically not correct. Please rephrase.

P2, line 52 and elsewhere, the term 'sero-diagnosis', 'sero-diagnostic' should be checked for consistency throughout the manuscript.

P2, line 53 and elsewhere, the term 'cross-reactivity' (with or without a hyphen) should be checked for consistency throughout the manuscript.

P3, line 71, 'Leishmania' should be in italics

P3, line 84, abbreviation of 'LTF' should be explained due to the first appearance in the text

P3, line 91, please add a comma after 'and'

P5, line 144, the source of BL21 (DE3) should be added to the manuscript

P5, line 146, should read 'B-cell'

P5, line 155, should read '10.000'

P5, line 163, please provide a reference or manufacturer specification of the monoclonal antibody
P6, line 183, '3' should be spell out in full
P6, line 193 'AA' should be spell out in full
P6, line 194, please delete 'of'
P8, line 254, please substitute 'recently' with 'previously'. The publication cited is from 2010 and 13 years ago and the term 'previously' would much better indicate that.
P8, line 255,to test an improved VL-specific ELISA.... Maybe 'evaluate' would be a better choice
P8, line 266, please refer to the respective figure
P8, line 267 and 268, statement requires a citation
P8, line 267, the term '8.3 TR' requires a short explanation, maybe in the introduction
P9, line 269, please refer to figures
P9, line 272, the authors stated that no cross-reactivity has been observed among the patient sera tested. Due to the small sample size, this statement is not justified by the data presented herein. Please rephrase sentence appropriately.

Reviewer #4 (Comments for the Author):

Abstract

Line 40: Replace infectious diseases' with 'vector-borne infections'.
Line 47: Remove infectious'

Materials and Methods

Line 109-110: The sentence 'Confirmation....' can be deleted as it is mentioned in lines 106-107.
Line 116: In addition to the names of diseases like malaria, and toxoplasma, mention the organisms like Plasmodium falciparum, Toxoplasma gondii, Trichinella spiralis, and so on. Mention genus and specie.
Line 152: BSA and RT. Write in full.
Line 176: Mention the distribution of the data (Gaussian vs non-Gaussian) to justify the statistical tools used

Results

Please incorporate the statistical tests in the results section along with the P-value. It is very informative to the reader to know the statistical basis for each result.
Line 194: Correct to 'a substitution of uncharged for uncharged'
Line 199: Change 'was' to 'were'.
Line 201: Change 'show' to 'showed'.
Line 209: How is this shown?
Line 229: Add 'Chagas disease (Trypanosoma cruzi, n=8)
Line 233: Change the sentence to indicate the absence of cross-reactivity by rKLi8.3 for T.cruzi and the presence of cross-reactivity of T. cruzi with Tecan/Novalisa.
Line 233: Change the sentence to indicate the absence of cross-reactivity by rKLi8.3 for T.cruzi and the presence of cross-reactivity of T. cruzi with Tecan/Novalisa.
Line 240: change 'show' to 'showed'

Discussion

Line 255: Remove the aim as it has been mentioned in the last paragraph of the introduction
Line 262: correct to ' substitution of charged for uncharged AA '
Line 270: Explain why rKLi8.3 is more sensitive and specific. Explanations for the cross-reaction of T. cruzi, the superiority of rKLi8.3 to TECAN/NovaLisa, etc. Simply explain the bunch of good results and briefly compare them with other studies.
Line 282: add a paragraph on the study limitations, if applicable.

References

Line 209: Reference 1 is issued by WHO. Please check.
Line 282: Reference 27 belongs to NIH. Please check.

Tables and Figures

Figures 1 and 2: D, Mention 'MA' in full.
Figure 4: In (A) add a color key on the image to indicate the three types of tests as in (B). Or it can be mentioned in the caption to cover both figures.
Table 1-4: Mention the unit of cut-off.
Table 5: Is the rKLi8.3 test already manufactured? or is it a trial product? Explain?

Staff Comments:

Preparing Revision Guidelines

Please return the manuscript within 60 days; if you cannot complete the modification within this time period, please contact me. If you do not wish to modify the manuscript and prefer to submit it to another journal, please notify me of your decision immediately so that the manuscript may be formally withdrawn from consideration by Microbiology Spectrum.

This paper describes the development of a novel more specific and sensitive ELISA and lateral flow test for the diagnosis of visceral Leishmaniasis in several countries to newly described antigen. The group compared previous antigens on a range of positive and control patients to understand the specificity and sensitivity.

Major Comments

It would be interesting to determine if the false positive seen in endemic controls in Fig 1 was positive in all of the tests/antigens or where these different FPs?

Do you see varying antibody response based on region of Sudan?

Fig 1, 2, 3, and 4.A - What statistic measure was used to designate your cut off or was this arbitrarily designated?

Does this have an effect on the sensitivity and specificity, what if you changed this to be the same for all antigens and compared values across the board for example all at 0.106?

Fig 1-3 - What comparison does the P value represent? One, two, three, or all compared?

Fig 2 – were co-infections considered in the positive TB samples?

Fig 3 – why do you switch up the colours of rKLi8.3 from green to blue? Please keep this consistent from Fig 1 and 2.

Fig 4 – please add colours in the figure description or onto the table, or alternatively add headings into the VL specific antibodies above each third. Change “Roc” to ROC.

All tables; if you had the same cut offs, would this impact sensitivity and specificity?

Please find the names of the various “anonymous” authors in the references.

Minor Comments

Line 45 – “L” in linked should be capitalized.

Line 59 – Cross reactivity should be two words or hyphenated.

Line 71 – infection should be plural “infections”.

Line 97 – remove “here”

Line 101 – please add references to where the previous ethics were approved for obtaining these samples from patients.

Line 163 – please name the control protein used.

Line 199 – add spaces between “L. and species”.

Supplemental figure 1 – “controll” is misspelled.

SF3 – Again, was the same false positive the same in all 3 antigen ELISA tests?

Manuscript Number: Spectrum04338-22

**Development of a novel ELISA and lateral flow test system for improved serodiagnosis of visceral Leishmaniasis in different endemic areas
(Prof Amer Al-Jawabreh)**

Abstract

Line 40: Replace infectious diseases' by vector-borne infections'.

Line 47: Remove infectious'

Materials and Methods

Line 109-110: The sentence 'Confirmation....' can be deleted as it is mention in lines 106-107.

Line 116: In addition to the names of diseases like malaria, toxoplasma mention the organisms like *Plasmodium falciparum*, *Toxoplasma gondii*, *Trichinella spiralis*, and so on. Mention genus and specie.

Line 152: BSA and RT. Write in full.

Line 176: Mention the distribution of the data (Gaussian vs non Gaussian) to justify the statistical tools used

Results

Please incorporate the statistical tests in results section along with P-value. It very informative to the reader to know the statistical basis for each result.

Line 194: Correct to ' a substitution of uncharged for uncharged'

Line 199: Change 'was' to 'were'.

Line 201: Change 'show' to 'showed'.

Line 209: How is this shown?

Line 229: Add 'Chagas disease (*Trypanosoma cruzi*, n=8)

Line 233: Change the sentence to indicate absence of cross reactivity by rKLi8.3 for *T. cruzi* and the presence of cross-reactivity of *T. cruzi* with Tecan/Novalisa.

Line 233: Change the sentence to indicate absence of cross reactivity by rKLi8.3 for *T. cruzi* and the presence of cross-reactivity of *T. cruzi* with Tecan/Novalisa.

Line 240: change 'show' to 'showed'

Discussion

Line 255: Remove the aim as it has been mentioned in the last paragraph of the introduction

Line 262: correct to ' substitution of charged for uncharged AA '.

Line 270: Explain why rKLi8.3 is more sensitive and specific. Explanations for the cross-reaction of *T. cruzi*, superiority of rKLi8.3 to TECAN/NovaLisa, etc. Simply explain the the bunch of good results and briefly compare with other studies.

Line 282: add a paragraph of the study limitations, if applicable.

References

Line 209: Reference 1 is issued by WHO. Please check.

Line 282: Reference 27 belongs to NIH. Please check.

Tables and Figures

Figure 1 and 2: D, Mention 'MA' in full.

Figure 4: In (A) add a color key on the image to indicate the three types of tests as in (B). Or it can be mentioned in the caption to cover both figures.

Table 1-4: Mention the unit of cut-off.

Table 5: Is the rKLi8.3 test already manufactured? or is it a trial product? Explain?

Supplementary files

Fig S1: Add the full name for the abbreviations: N, C, T, C, AB, S

Figure S3: Change 'toxoplasmosis' to '*Toxoplasma gondii*'

Change 'malaria' to '*Plasmodium spp.*'

Point by point reply

We thank both reviewers for taking time and their constructive comments which improved our manuscript. Thank you very much!

Please find below our point by point reply:

Reviewer #3

First, we want to thank for the appreciation of our work!

- 1) *the number of serum samples from patients with other bacterial or parasitic infections used as further controls is very small...*

Reply: We agree that the number of sera with other infections is limited. However, it is extremely difficult to get enough control sera from patients with other infections from the same (VL) endemic regions. Despite these limitations, no cross-reactivity was observed with 11 different pathogens. In the revised manuscript, we mentioned the difficulties to obtain control sera (line 279-282).

- 2) *In their previous work, they used a recombinant antigen, rKLO3 from an L. donovani isolate. Here, a novel antigen from L. infantum has been used for evaluation and optimization of the test. It is not entirely clear, why the authors decided to use a kinesin from L. infantum. Also, the rationale why a comparative sequence analyses of kinesins of L. infantum strains were conducted is not clear as well (see page 6, lines 191-196). Why they end up with the three kinesins described in the following paragraph? This should be clarified in the result.*

Reply: The aim of the project was to identify and develop an antigen that provides better diagnostic performance in East-Africa than rKLO8 or the rK39 from *L. donovani*. Our selection was based on the following rationale: Antigens that contain charged AA give better antibody responses than antigens with uncharged AA, i.e. the substitution of charged for uncharged AA reduces the antigenicity. We thus compared multiple recombinant kinesin proteins from different strains (*L. donovani*, *L. archibaldi* and *L. infantum*). We identified only one kinesin protein (rKLi8.3 from *L. infantum*) where charged AA were not substituted (Supplemental Table S2). Besides the sequence, the structure of the kinesin antigen (number of tandem repeats) might also influence the antigenicity, as predicted by BepiPred and shown in supplemental figure S3. Since these analyses were only in silico-predictions, we therefore tested the influence of kinesin sequence (Fig 1) and structure (Fig 2) on antibody binding by ELISA. The results demonstrated that the performance of kinesin from *L. infantum* with 8.3 tandem repeats (rKLi8.3) was superior to *L. donovani* (rKLd8.3) and *L. archibaldi* (rKLa8.3). To assess whether rKLi8.3 improves VL sero-diagnosis, we compared the performance of rKLi8.3 with our previously developed rKLO8 and the commercially available rK39 in ELISA. This is now explained in line 213-215 in the revised version.

3) *To show the differences between the three selected antigens (rKLi6.3, rKLi7.3, and rKLi8.3) and to make the manuscript more reader-friendly, a separate figure showing the structural differences of these three antigens would be strongly recommend.*

Reply: In the revised version, we have added a scheme (new supplemental figure S2), which illustrates the structures of rKLi6.3, rKLi7.3 and rKLi8.3.

4) *In figure 1, the authors presented data from the ELISA performed with three different antigens. It seems that the identical data collected with the rKLi8.3 antigen were also used for figure 2C and 3C. The authors should clearly indicate whether these data were used for the calculations of the data presented in figure 2D and 3D. To avoid any misunderstandings, the data obtained with rKLi8.3 should be only presented in figure 1.*

Reply.: We apologize for our mistake in showing the same rKLi8.3 data in Figs 1, 2 and 3. As we have always tested the different antigens simultaneously with rKLi8.3, we have included the corresponding rKLi8.3 data in the revised version for Figs 1, 2 and 3. We also used the new data for the calculation.

5) *On page 7, second paragraph, to evaluate the performance of the ELISA, two additional antigens, rK39 and rKLO8, were investigated. The origin or source of these antigens are unclear. Should the antigens genetically engineered by the authors, a full description including construction (amplification, cloning, etc.) and purification have to be included in the Materials and Methods. Should these antigens are commercially available, the source of these proteins have to be included.*

Reply: Thank you very much for reminding us to refer to the origin of rK39 and rKLO8.

We inserted a new paragraph in the Material and Method section, explaining the origin and cloning of both proteins, line 148-151. The rK39 protein was purchased from *Rekom. Biotech, S.L., Granada Spain* and Cloning; expression and purification of rKLO8 has recently been described by us, *Plos Neglected tropical disease*, 18, 2013.

6) *The discussion is very short and needs improvement. A comprehensive discussion of the strengths and limitation (e.g. sample size of the controls are very small) of the study are strongly recommended as well as the future perspectives how to increase the sensitivity of the novel test systems. Also, the benefit of the developed tests described herein compared to in vitro diagnostics currently used have to be explored in a better way.*

Reply: We have improved the discussion by more explicit mentioning the strength and limitations of the newly developed diagnostic tests, line 279- 291.

7) *It is not clear to what the number of replicates of each serum sample was used in the ELISAs and LFT. This needs clarification*

Reply: All ELISAs has been repeated twice with serum samples in duplicates. According to the manufactures protocols, valid LF test (strong internal control band) were performed once. If the control band was weak or absent, LF tests were repeated. This information is now included in the Material and Method section.

8) *The definition of the cut off values should be explained more clearly. It is not entirely clear why the authors define the mean absorbance values 'plus three standard deviations'. Please clarify.*

Reply: The cut-off values determine whether a serum sample is considered positive or negative. Determination of the cut-off value is a standard procedure which is calculated as follows: Mean OD value of healthy controls (\bar{x}) + 3 standard deviation (**SD**). OD values above $\bar{x}+3SD$ were considered positive. To achieve a most accurate cut-off value, **50** negative sera were used. Furthermore, the cut-off value for each recombinant protein was confirmed using the receiver operating characteristic (ROC) curve. This has been mentioned in statistical analysis.

9) *Minor points*

Reply: All minor points are directly addressed in the manuscript.

Reviewer #4

First, we also want to thank the reviewer 4 for very careful reading and commenting our work!

If not explicitly mentioned in this letter, all minor points have been directly addressed in the revised manuscript according to the suggestions of the reviewer.

Line 183: Mention the distribution of the data (Gaussian vs. non-Gaussian) to justify the statistical tools used.

Reply: Comparing defined patient entities (VL vs. endemic controls; VL vs Malaria; VL vs Tuberculosis; etc.) we compare 2 or more independent groups. The data of the antibody response of defined patient group clearly show a parametric (Gaussian) distribution. Therefore we employed student's t-test for two independent groups or one way ANOVA for several independent groups. In all cases (comparison of two or multiple groups) the

significance was $P < 0.0001$. This is now explained in the revised version, section statistical analysis.

Line 270: Explain why rKLi8.3 is more sensitive and specific. Explanations for the cross-reaction of T. cruzi, the superiority of rKLi8.3 to TECAN/NovoLisa, etc. Simply explain the bunch of good results and briefly compare them with other studies.

Reply: The rKLi8.3 prototype ELISA exceeded TECAN/NovoLisa in terms of sensitivity and specificity. Sera from malaria, tuberculosis and Chagas patients have been shown to cross-react with tests using rK39 or whole parasite lysate. This may be due to shared antigenic determinants or a close phylogenetic relationship of e.g. *Leishmania* with Trypanosomes, which both belong to the family of kinetoplastida. In contrast, the rKLi8.3 ELISA and LFT showed no cross-reactivity with sera from tuberculosis-, malaria- and Chagas patients. The high specificity of rKLi8.3 is most likely due to the combination of a conserved kinesin sequence with high antigenicity (charged AA) and the increased number of TRs. This is now included in the discussion of the revised version, line 281-291.

Table 5: Is the rKLi8.3 test already manufactured? Or is this a trial product? Explain

Reply: The LFT and ELISAs have been established and manufactured but not yet released on the market. As they are not yet commercially available, we still call them prototype test.

We hope that the revised manuscript is now acceptable for publication in Spectrum Microbiology.

Best regards,

Ulrich Steinhoff

March 16, 2023

Prof. Ulrich Steinhoff
Philipps Universität Marburg
Medical Microbiology and Hospital Hygiene
Marburg
Germany

Re: Spectrum04338-22R1 (Development of a novel ELISA and lateral flow test system for improved serodiagnosis of visceral Leishmaniasis in different endemic areas)

Dear Prof. Ulrich Steinhoff:

Thank you for submitting your manuscript to Microbiology Spectrum. As you will see your paper is very close to acceptance. Please modify the manuscript along the lines I have recommended. As these revisions are quite minor, I expect that you should be able to turn in the revised paper in less than 30 days, if not sooner. If your manuscript was reviewed, you will find the reviewers' comments below.

When submitting the revised version of your paper, please provide (1) point-by-point responses to the issues raised by the reviewers as file type "Response to Reviewers," not in your cover letter, and (2) a PDF file that indicates the changes from the original submission (by highlighting or underlining the changes) as file type "Marked Up Manuscript - For Review Only". Please use this link to submit your revised manuscript. Detailed instructions on submitting your revised paper are below.

Link Not Available

Sincerely,

Kileen Shier

Editor comments:

Line 76: In your Introduction, remind readers which species are associated with visceral leishmaniasis.

Line 180-182: What manufacturer? No manufacturer is mentioned in this section other than the dispense platform manufacturer.

Line 244: Spell out DEV

Line 420: Are these variations species-specific?

Figure 4: Was there statistical significance between the VL and CL groups, and between the VL and Chagas groups? If this makes the figure too complicated, consider grouping the three CL species into one group.

Preparing Revision Guidelines

- Point-by-point responses to the issues raised by the reviewers in a file named "Response to Reviewers," NOT IN YOUR COVER LETTER.

- Upload a compare copy of the manuscript (without figures) as a "Marked-Up Manuscript" file.
- Each figure must be uploaded as a separate file, and any multipanel figures must be assembled into one file.
- Manuscript: A .DOC version of the revised manuscript
- Figures: Editable, high-resolution, individual figure files are required at revision, TIFF or EPS files are preferred

Please return the manuscript within 60 days; if you cannot complete the modification within this time period, please contact me. If you do not wish to modify the manuscript and prefer to submit it to another journal, please notify me of your decision immediately so that the manuscript may be formally withdrawn from consideration by Microbiology Spectrum.

To the editor

Dr. Kileen Shier

Microbiology Spectrum

Fachbereich Medizin

Institut für Medizinische Mikrobiologie
und Krankenhaushygiene

Prof. Dr. Ulrich Steinhoff

Tel.: +49 (0)6421 / 28 - 66134

Fax: +49 (0)6421 / 28 - 64344

E-Mail: ulrich.steinhoff@staff.uni-marburg.de

Sek.: Melanie Wolf

Tel.: 06421 / 58 - 66455

E-Mail: wolfme@med.uni-marburg.de

Adresse: BMFZ, Hans-Meerwein-Str.2
35043 Marburg/Germany

Web: www.uni-marburg.de

Marburg, 21.03.2023

Dear Dr. Shier,

we thank you for the comments which further enhanced the clarity and significance of our study. According your suggestions, we addressed all points and marked the changes in the revised version of the manuscript.

- The Leishmania species causing VL have now be named in the introduction (line 71/72).
- The manufacturer (Eurofins-Ingenasa) is now named in the Method section (line 175).
- DEV has been spelled out (line 422).
- We have clarified that the observed AA variations were associated with the origin of the parasite and not necessarily with the *Leishmania* species (line 199).
- We performed the statistical significance between VL and CL and Chagas in figure 4, please see below. However, mathematical significance analysis might lead to the false interpretation that TECAN and NovaLisa – ELISAs are giving none-valid results. In contrast to the rKLi8.3 ELISA, TECAN and NovaLisa are less sensitive and crossreact with Chagas sera, but are still able to detect VL patients, as determined by the values exceeding the cut off. Although the statistics would be in favour of our rKLi.8.3 ELISA, we strongly suggest to use the Fig. 4 without statistics, as in the case of the ELISA-interpretation statistical analysis and biological interpretation do not fully correlate.

Thank you for the efficient handling and the support of the manuscript and hope that our MS is now acceptable for publication in spectrum microbiology.

Best regards,

Fig 4 new for the editor

Figure 4. Assessment of VL-specific antibodies by (A) rKLi8.3 prototype ELISA, (B) commercial-TECAN - and (C) NovaLisa ELISA-kits. Sera from VL-patients infected with *L. donovani*/*L. infantum* [n=8], CL-patients infected with *L. tropica* [n=6], *L. major* [n=6], *L. brasiliensis* [n=2] and Chagas-disease patients infected with *T. cruzi* [n=8] were tested. Manufacturer-defined borderline zone is indicated by red lines; data within were considered negative. (D) Roc curve analysis of TECAN, NovaLisa and rKLi8.3 prototype ELISA. Statistical analysis was performed by One-way ANOVA (Graphpad Prism) with p-values shown. $p < 0,01$ (*), $p < 0,001$ (**) and $p < 0,0001$ (***).

March 24, 2023

Prof. Ulrich Steinhoff
Philipps Universität Marburg
Medical Microbiology and Hospital Hygiene
Marburg
Germany

Re: Spectrum04338-22R2 (Development of a novel ELISA and lateral flow test system for improved serodiagnosis of visceral Leishmaniasis in different endemic areas)

Dear Prof. Ulrich Steinhoff:

Your manuscript has been accepted, and I am forwarding it to the ASM Journals Department for publication. You will be notified when your proofs are ready to be viewed.

Sincerely,

Kileen Shier
Editor, Microbiology Spectrum
